# Biological Functions of HMGN Chromosomal Proteins

**DOI:** 10.3390/ijms21020449

**Published:** 2020-01-10

**Authors:** Ravikanth Nanduri, Takashi Furusawa, Michael Bustin

**Affiliations:** Laboratory of Metabolism, Center for Cancer Research, National Cancer Institute, National Institutes of Health, Bethesda, MD 20892, USA; ravikanth.nanduri@nih.gov (R.N.); takashi.furusawa@nih.gov (T.F.)

**Keywords:** HMGN proteins, chromatin, epigenetics, gene regulation, development

## Abstract

Chromatin plays a key role in regulating gene expression programs necessary for the orderly progress of development and for preventing changes in cell identity that can lead to disease. The high mobility group N (HMGN) is a family of nucleosome binding proteins that preferentially binds to chromatin regulatory sites including enhancers and promoters. HMGN proteins are ubiquitously expressed in all vertebrate cells potentially affecting chromatin function and epigenetic regulation in multiple cell types. Here, we review studies aimed at elucidating the biological function of HMGN proteins, focusing on their possible role in vertebrate development and the etiology of disease. The data indicate that changes in HMGN levels lead to cell type-specific phenotypes, suggesting that HMGN optimize epigenetic processes necessary for maintaining cell identity and for proper execution of specific cellular functions. This manuscript contains tables that can be used as a comprehensive resource for all the English written manuscripts describing research aimed at elucidating the biological function of the HMGN protein family.

## 1. Introduction

High mobility group N (HMGN) is a family of non-histone chromatin architectural proteins, ubiquitously expressed in the nuclei of vertebrate cells [1,2] that binds specifically to nucleosomes, the building block of the chromatin fiber. HMGNs bind to nucleosomes dynamically without any specificity for DNA sequence [3,4,5]. The interaction of HMGNs with nucleosomes weakens the binding of linker histone H1 to nucleosomes, thereby affecting the local and global chromatin structure [6,7,8,9]. Genome-wide HMGNs preferentially localize to chromatin regulatory sites such as enhancers and promoters, thereby affecting the integrity of the chromatin epigenetic landscape and the cellular transcription profile [10,11].

The HMGN family consists of five proteins named HMGN1, 2, 3, 4 and 5, all containing a bipartite nuclear localization signal, a negatively charged chromatin regulatory domain located at C terminal region of the protein, and a positively charged nucleosome binding domain, which is highly conserved and is the consensus sequence of this protein family [5]. The major members of this family, HMGN1 and HMGN2, have been detected in every vertebrate tissue examined. The expression of HMGN3, 4 and 5 proteins seems to be more restricted and with a few notable exceptions [12,13], the level of these HMGN variants is significantly lower than that of HMGN1 and HMGN2. The genes coding for HMGN1, 2, 3 and 5 have a very similar intron-exon organization, raising the possibility that they originated from a common ancestor. The gene coding for HMGN4 is the product of a retroviral transposon and appeared to be restricted to primates [2,14]. Both the human and mouse genome contains multiple *HMGN1* and *HMGN2* retropseudogenes dispersed throughout their genome [15,16]; however, to date, only *HMGN4* has been identified as a retropseudogene coding for a protein. In humans, *HMGN1* is located on chromosome 21 (21q22.3), *HMGN2* is located on Chromosome 1 (1p36.1), *HMGN3* and *HMGN4* are located on chromosome 6 (6q14.1 and 6p21.3) and *HMGN5* is located in Chromosome X (Xp13.3) [17]. HMGN1-4 have similar molecular weights, around 10 kDa (∼90 amino acids length), while HMGN5 contains a long C-terminus; the length of this region varies among species [14,17].

Given the ubiquitous expression of HMGNs in vertebrate cells, their specific binding to nucleosomes, and their preferential association with chromatin regulatory sites it could be expected that these proteins would affect cell type-specific gene expression programs. Indeed, it has been reported that HMGNs play a role in embryogenesis and affect neuronal, ocular, reproductive, and pancreatic cell differentiation. In addition, several studies have suggested that HMGNs play a role in DNA repair processes and that the loss of HMGNs can lead to cancer, neurological disorders, and altered immune functions.

Likely, HMGNs affect the cellular phenotype by modulating epigenetic processes that affect cell type-specific gene expression [11,18], a subject of considerable interest that is not discussed here. In this review, we focus on the biological function of HMGN and summarize the available data on the role of HMGN proteins in developmental processes and in disease etiology. The two tables included in the manuscript provide a comprehensive resource for references to all the English written manuscripts on these topics that have been published in PubMed (https://www.ncbi.nlm.nih.gov/pubmed) up to December 2019.

## 2. Genetically Altered HMGN Mice

A major approach used to study the biological function of HMGN involves correlative studies in which changes in cellular phenotypes are linked to changes in the expression level of specific HMGN variants. A second important approach involves analysis of cells and tissues in which the levels of HMGNs are experimentally altered by using specific vectors to either increase, decrease, or fully delete the expression of a specific HMGN variant. In this approach, the use of genetically altered mice can provide information on the function of HMGN variants at the biological level of an entire organism.

Table 1 lists the genetically altered HMGN mice that have been used to study HMGN function. The phenotype of most of these lines has also been analyzed in detail by the German Mouse Clinic and is posted online at https://www.mouseclinic.de. With one exception [19] the results indicate that mice lacking HMGN variants are born, appear normal and survive; however, each of the genetically altered mouse line shows phenotypic differences from its wild-type littermate mouse line, especially when exposed to stress.

## 3. HMGN Proteins in Development

### 3.1. Embryogenesis

HMGN expression is developmentally regulated; significant HMGN1 and 2 expression is seen during mouse oogenesis and early preimplantation [20,21]; however, during subsequent development and differentiation, the expression levels of these variants decreases [22,23]. Quantitative RNA blot analysis of whole mouse embryos indicates that during differentiation from E7 to E17, the relative levels of HMGN transcripts decrease by 50%, and that the relative levels of HMGN transcripts in the adult mouse heart or spleen are only 5% of those detected in 7-day embryos [22]. Nevertheless, every tissue examined in the adult mouse contains *Hmgn1,2* transcripts and HMGN proteins. The high relative levels of HMGN expression during early embryogenesis suggests that these proteins may play a role in lineage commitment and cellular differentiation.

Indeed, several findings suggested that proper expression level of HMGN1 and HMGN2 may play a role during early stage embryogenesis. Thus, prominent expression of HMGN1 and HMGN2 is seen in mouse embryonic stem cells (ESCs) [20], in induced pluripotent stem cells (iPSCs) [11], and in adult mouse hair follicle bulge where hair stem cells are located [24]. *Hmgn1* and *Hmgn2* transcripts are detected during mouse oogenesis, and after fertilization, maternal *Hmgn* transcripts accumulate in the two-cell stage embryo. Transient depletion of HMGN1 and HMGN2 proteins by injecting antisense oligonucleotides into mouse oocytes, delayed cell cleavage and the onset of the blastocyst stage [21]. Both HMGN1 and HMNG2 are highly expressed throughout mouse preimplantation stages, including blastocyst stage, and both HMGN1 and HMGN2 are strongly expressed in ESCs [20,25]. In tissue culture, loss of HMGNs does not affect the rate of differentiation of ESCs into embryoid bodies; however, the gene expression profile of the *Hmgn^−/−^* cells was clearly different from that of cells obtained from genetically matched WT littermates [20]. A recent study shows that CRISPR-mediated knock out of HMGN2 in P19, a pluripotent stem cell model, alters the epigenetic landscape of these cells, leading to changes in gene expression. The authors conclude that HMGN2, and to a lesser degree, HMGN1, stabilize the epigenetic landscape necessary for maintaining the pluripotent identity of these cells [26].

Additional support that HMGNs may play a role in early development is seen in the initial reports on the *Hmgn1^−/−^* mouse line which indicate that the frequency ratio of *Hmgn1^−/−^* pups resulting from mating of *Hmgn1^+/−^* heterozygotes was 0.08 rather than 0.25, as would be expected from a Mendelian distribution of the *Hmgn1^−/−^* allele. The low number of *Hmgn1^−/−^* offspring seems to be due to events occurring in early stages of embryonic development, since the genotype distribution in 11.5-day embryos was the same as that seen in born pups. In addition, the average litter size obtained from mating of *Hmgn1^−/−^* was approximately 30% lower than that resulting from *Hmgn1^+/+^* mating [27].

In mice, downregulation of both *Hmgn1* and *Hmgn2* transcripts is seen during chondrocyte, kidney, hair follicle, and corneal development and differentiation [24,28,29]. In addition, cell culture studies show downregulation of HMGN1 and HMGN2 expression during myogenesis, erythropoiesis and chondrogenesis [28,30,31]. In these tissue culture experiments, aberrant expression of HMGN proteins inhibits myoblast and chondrocyte differentiation. It may be relevant that studies by the German Mouse Clinic (https://www.mouseclinic.de) indicate that the average size of the double knock-out mice lacking both HMGN1 and HMGN2 is smaller than that of wild-type (WT) littermate mice, a finding also observed by our laboratory.

During *Xenopus* development, *Hmgn1* and *Hmgn2* transcripts are detected in the three *Xenopus* germinal layers; however, in the embryos, HMGN1 and HMGN2 proteins are detected and accumulate only after mid blastula transition (MBT). Morpholino mediated HMGN1 and HMGN2 depletion, or vector mediated HMGN overexpression prior to MBT does not affect *Xenopus* development; however, after MBT, aberrant HMGN expressions cause imperfectly closed blastopores, distorted body axis, and embryos with abnormal head structures [29]. These findings suggest that in *Xenopus*, correct HMGN expression is crucial for proper development. The differences between *Xenopus* and mice in the effect of HMGNs on differentiation raises the possibility that the requirement for HMGN functions are more stringent in “lower” vertebrates; perhaps “higher” vertebrates developed redundant mechanisms to bypass the absolute requirement for proper expression of HMGN variants.

Chromatin is not only a major regulator of gene expression but seems to play a structural role in supporting the mechanical properties of the nucleus [32]. In this aspect, overexpression of HMGN5 variant has been shown to decrease nuclear sturdiness, most likely by reducing the binding of histone H1 to nucleosomes [7], thereby leading to chromatin decompaction. In agreement with this possibility, transgenic mice overexpressing HMGN5 develop hypertrophic heart with large cardiomyocytes, deformed nuclei and disrupted lamina and die of cardiac malfunction [33]. Mice lacking HMGN5 show several phenotypes (https://www.mouseclinic.de) including changes in transcription of genes that alter glutathione metabolism [34].

### 3.2. Neuronal Development

The database in the Allen Institute of Brain Sciences (https://alleninstitute.org/) show high HMGN expression in both human and mouse brain, especially in the eye, hippocampus, and SVZ regions. In mouse ESCs, loss of HMGN1 and HMGN2 reduces the expression of OLIG1 and OLIG2, two transcription factors known to play an important role in oligodendrocyte differentiation. Indeed, these *Hmgn1^−/−^Hmgn2^−/−^* (DKO) ESCs showed a reduced ability to differentiate towards the oligodendrocyte lineage and mice lacking these HMGNs showed reduced oligodendrocyte count, decreased spinal cord myelination, and display several neurological phenotypes that could be related to the decreased oligodendrocyte count and faulty nerve myelination [20]. A recent study using a new line of *Hmgn2^−/−^* mice reports that mice lacking HMGN2 show reduced cortical surface and microcephaly, suggesting a role for HMGN2 in corticogenesis. These *Hmgn2^−/−^* mice were sub viable and most die within a short time after birth [19] an unexpected observation since previous studies showed that mice lacking either HMGN1 or HMGN2, or even both these variants survive [10]. The differences may be due to the targeting strategy and the exact timing of HMGN deletion during development.

Cell culture studies [35] revealed that forced expression of HMGNs increased the generation of astrocytes, while knockdown of HMGNs in neural progenitor cells suppressed astrocyte differentiation. Most likely, HMGNs affect astrocyte differentiation by modulating the JAK-STAT pathway. HMGN3 is expressed in several regions of adult mouse brain that are enriched in astrocytes [36]. In addition, HMGN3 regulates GLYT1 expression, a protein known to plays an essential role at glycinergic and glutamatergic synapses in the brain and central nervous system [37,38]. In the mouse neuroblastoma cell line N1E-115, the mRNA encoding HMGN5 localizes to growth cones of both neuron-like cells and of hippocampal neurons, where it has the potential to be translated, and that HMGN5 can be retrogradely transported into the nucleus along neurites. Loss of HMGN5 function induces transcriptional changes and impairs neurite outgrowth, while HMGN5 overexpression induces neurite outgrowth and chromatin decompaction suggesting that this HMGN variant plays a role in facilitating growth cone-to-nucleus signaling during neuronal development [39].

The human *HMGN1* gene is located in chromosome 21 [40] in a segment known as the Down Syndrome critical region (DSCR), which is thought to play an important role in the etiology of this syndrome. Elevated HMGN1 expression is observed in human Down Syndrome cells [41] and in cells from Ts1Cje mouse, which is used as a model for Down Syndrome [42]. HMGN1 was shown to affect the expression of the methyl CpG-binding protein 2 (MeCP2) a DNA-binding protein known to affect neurological functions including autism spectrum disorders. Quantitative PCR and Western analyses of cell lines and brain tissues from mice that either overexpress or lack HMGN1 indicate that HMGN1 is a negative regulator of MeCP2 expression [43]. Alterations in HMGN1 levels lead to changes in chromatin structure and histone modifications in the MeCP2 promoter. Behavior analyses by open field test, elevated plus maze, reciprocal social interaction, and automated sociability test, link changes in HMGN1 levels to abnormalities in activity and anxiety and to social deficits in mice.

Taken together, the available findings support the possibility that aberrant expression of HMGN variants can lead to neurodevelopmental disorders.

## 4. Ocular Development

A systematic investigation of the expression patterns of HMGN1, HMGN2 and HMGN3 in the mouse eye development revealed that each HMGN variant is expressed in a specific manner [44]. At embryonic day 10.5 (E10.5) HMGN1 and HMGN2 proteins are evenly distributed in all ocular structures while HMGN3 expression is not detected at this stage. At E13.5, a decrease in HMGN1, but not in HMGN2 protein level is detected in the newly formed lens fiber cells. At this stage, HMGN3 is detected in the developing corneal epithelium and lens fiber cells. In adult mice, HMGN1 and HMGN2 are detected throughout the retina and lens, but in the cornea, these HMGNs are predominately located in the epithelium. In the cornea, HMGN3 is transiently upregulated in the stroma and endothelium at birth, while in adults, HMGN3 expression is restricted to the corneal epithelium. In the retina, HMGN3 expression is upregulated around 2 weeks of age and remains at relatively high levels in the inner nuclear and ganglion of the adult retina. Thus, even though all members of the HMGN protein family have similar protein structures, in the eye, each protein has a distinct regional and developmental stage expression pattern. In *Hmgn1^−/−^* mice the corneal epithelium is thin, has a reduced number of cells, is poorly stratified and shows additional abnormalities, all suggesting that loss of HMGN1 leads to transcriptional changes that ultimately affect corneal development and maturation [28].

## 5. Reproductive System

HMGN5 is highly expressed in placental rather than embryonic tissues, and is highly enriched in trophoblast giant cells, which play a key role in placental differentiation. In Rcho-1 cells, a cell line that serves as a model system for studies on the development of trophoblast giant cells, the expression levels of HMGN5 is highly correlated with prolactin genes and placental lactogens [12]. *Hmgn3* is highly expressed in the decidua and decidualizing stromal cells or artificially stimulated decidualization leads to *Hmgn3* expression. In vitro, overexpression of HMGN3 in uterine stromal cells enhanced the expression of decidualization markers, whereas inhibition of HMGN3 reduced their expression [45]. In sum, the data show a correlation between expression of specific HMGN variants in the reproductive system cells; however, it is still not clear whether proper HMGN expression in these cells is necessary for their adequate function.

## 6. Pancreas

In both the human and mouse pancreas, HMGN3 is specifically expressed in adult pancreatic islet cells [13]. All the endocrine cell types including the insulin-producing beta-cells and glucagon-producing alpha cells express high levels of HMGN3 [46]. *Hmgn3^−/−^* mice are viable and appear normal; however, compared to their wild-type littermates, their blood glucose levels are elevated, their serum insulin levels are reduced, and their glucose tolerance is impaired. In vitro studies using the mouse insulinoma MIN6 cell line revealed that HMGN3 modulates the expression of several genes involved in insulin secretion, including that for the glucose transporter GLUT2. In both MIN6 cells and *Hmgn3*^−/−^ mice, the loss of HMGN3 protein reduces the levels of GLUT2. HMGN3, but not the HMGN1 or HMGN2 variants, binds specifically to chromatin in the promoter region of the *Glut2* gene and enhances its expression by facilitating the recruitment of PDX1 and additional transcription factors, to the *Glut2* promoter. Chromatin immunoprecipitation (ChIP) experiments reveal that HMGN3 enhances the acetylation of H3 at the *Glut2* promoter and suggest that HMGN3 and PDX1 mutually reinforce their interaction with the chromatin at the *Glut2* promoter but not throughout the chromatin or in the nucleoplasm.

In *Hmgn3^−/−^* mice, loss of HMGN3 reduces the levels of glucagon in the plasma. However, analysis of TC1-9 cells, a cell line that is widely used for studies on pancreatic alpha-cell function, did not reveal a major role for HMGN3 in glucagon synthesis and secretion. These results indicate that HMGN3 reduces the levels of glucagon in the plasma of mice without affecting glucagon gene expression [46].

Table 2 lists the references accessible in PubMed (https://www.ncbi.nlm.nih.gov/pubmed) discussing the role of HMGN variants in various developmental processes. The references are limited to articles written in English. Taken together, the available data suggest that mice lacking HMGN genes are born and survive; however, loss of HMGN occasionally leads to reduced litter size, to smaller sized mice, and to epigenetic changes that play a role in the etiology of several developmental disorders.

## 7. HMGN Proteins in Disease

The widespread occurrence of HMGN variants in all vertebrate cells, the high degree of sequence conservation of the proteins among all vertebrates, and the localization of the proteins to chromatin regulatory sites, suggest that they are required for proper cellular function. Table 3 lists the references describing investigations of the biological function of HMGNs, especially their possible role in the etiology of disease. In addition, as described above, several mice lines that have been genetically modified to alter HMGN expression have been extensively phenotyped by the German Mouse Clinic (https://www.mouseclinic.de/); the results of these analyses are available online. Several general conclusions can be drawn from these studies: (1) Cells and mice with altered HMGN expression survive. (2) Invariably, altered levels of HMGN lead to changes in gene expression. (3) Loss or altered expression of HMGNs leads to both up and downregulation of tissue-specific gene expression. (4) HMGNs modulate the existing cell type-specific gene expression program. (5) Mice lacking HMGN show numerous phenotypes; however, the phenotypes are not strong. Part of the phenotypes seen suggest a possible role for HMGNs in cancer, neurological disorders and in immunological processes.

## 8. Cancer

Cancer initiation and progression often show changes in gene expression that reflect alteration in cell identity. Significantly, recent results show that HMGN1 and HMGN2 preferentially bind to cell type-specific chromatin regulatory regions and that loss of HMGN enhances the rate of transcription factor-induced reprogramming of mouse embryonic fibroblasts (MEF) into pluripotent cells, and the lineage conversion of MEFs into neuronal cells. These and additional experiments suggest that HMGN stabilizes cell identity, most likely by facilitating the binding of cell type-specific regulatory factors to cell type-specific regulatory sites [11]. A PubMed search on the role of HMGN proteins in cancer retrieved a wide range of studies involving analysis of human cell lines, mice models, and patient clinical data (Table 3). Comparative gene expression analyses of normal and cancer cells frequently show differences in HMGN expression. However, it is not yet clear whether the changes in HMGN expression levels are a cause or a consequence of the transcriptional changes seen in cancer cells.

High cytoplasmic expression and secretion of HMGN1 were associated with increased peritumor infiltration of lymphocytes in breast cancer [47,48]. In non-small cell lung cancer (NSCLC) patients, the levels of HMGN1 in the serum were correlated with overall survival after curative pneumonectomy. Likewise, the levels of HMGN1 in the serum of NSCLC individuals were higher in non-metastatic stages (I-III) as compared to the metastatic stage (IV), suggesting that HMGN1 can serve as a biomarker for early stages of NSCLC [49]. Leukemic B cells from patients with chronic lymphoblastic leukemia express HMGN2 on their surface; the HMGN2 variant acts as an autoantigen and contributes to the development of autoimmune hemolytic anemia [50]. Individuals with Down syndrome have a significantly increased risk of developing B cell acute lymphoblastic leukemia (B-ALL). HMGN1, coded by a gene located on human chromosome 21, seems to play a crucial role in the B-ALL. Studies with Ts1Rhr mice which contain a triplication of a segment of mouse chromosome 16 that is orthologous to human chromosome 21 and serve as models for Down syndrome, and analysis of cells derived from a transgenic mice overexpressing human HMGN1 [51] provide evidence that HMGN1 overexpression promotes B-ALL. In B-ALL, elevated levels of HMGN1 reduce the levels of H3K27me3, a histone modification associated with gene silencing, and increase the levels of H3K27ac, a modification associated with increased gene expression, thereby leading to transcriptional amplification and promoting B cell proliferation [52,53].

HMGN2 was also shown to promote breast cancer growth in response to prolactin. In this system, HMGN2 reduces the binding of linker histone H1 to chromatin regulatory regions, thereby enhancing STAT5 accessibility to promoter DNA, ultimately leading to increased proliferation of breast cancer cells [54,55]. Further analysis showed that the K2 residue of HMGN2 was deacetylated in primary breast tumors but highly acetylated in normal human breast tissue, implying that targeting HMGN2 deacetylation may be a viable treatment for this type of breast cancer [55].

The HMGN4 variant which is transcribed from a processed retropseudogene [56], is expressed only in humans. Upregulation of HMGN4 expression in mouse and human cells, and in the thyroid of transgenic mice, alters the cellular transcription profile, downregulates the expression of the tumor suppressors Atm, Atrx, and Brca2, and elevates the levels of the DNA damage marker *γ*H2AX. Mouse and human cells overexpressing HMGN4 show increased tumorigenicity as measured by colony formation, by tumor generation in nude mice, and by the formation of preneoplastic lesions in the thyroid of transgenic mice. In patients with hepatocellular carcinoma (HCC), elevated level of HMGN4 expression is associated with high grade tumors and shorter overall survival, suggesting that in patients with HCC, HMGN4 could act as a candidate biomarker for poor prognosis. Thus, HMGN4 may serve as an additional diagnostic marker or perhaps even a therapeutic target in certain thyroid cancers [56]. HMGN5 expression seems to be elevated in breast, prostate and bladder cancer cell lines (Table 3); however, the oncogenic potential of this variant has not yet been studied in detail.

In mice, HMGN1 was shown to bind to the promoters of the proto-oncogene *FosB* and *Hmgn1^−/−^* mice show upregulated expression of the proto-oncogenes *FosB*, *C-fos*, *BclL3* and *N-cadherin* and downregulation of *JunB*, and *c-Jun,* suggesting a possible role of this HMGN variant in modulating the action of these factor during cancer development [57,58]. Following a single DEN injection, mice lacking HMGN1 displayed earlier signs of liver tumorigenesis as compared to wild-type mice [59] suggesting that loss of HMGN1 leads to epigenetic changes that affect the rate of N-nitrosodiethylamine-induced hepatocarcinogenesis in mice. In support of this possibility, transformed *Hmgn1^−/−^* fibroblasts grow in soft agar and produce tumors in nude mice with significantly higher efficiency than WT littermate fibroblasts. *Hmgn1^−/−^* mice also show an increased incidence of tumors and metastases as compared to their *Hmgn1^+/+^* littermates [60]. Together, these results support the general notion that the presence of HMGN stabilizes the cell epigenetic landscape that regulates cell type-specific transcription and determines cell identity [11].

It is well established that faulty repair of damaged DNA is a major factor contributing to cancer development. Several types of experiments suggest a role for HMGN proteins in DNA repair processes [61,62]. Thus, exposure of the shaved backs of *Hmgn1^−/−^* mice and their *Hmgn1^+/+^* littermates to UV-B produced significant damage in the skin of *Hmgn1^−/−^* mice but not in the skin of control, wild-type littermates. Furthermore, mouse embryonic fibroblasts isolated from *Hmgn1^−/−^* mice were significantly more sensitive to UV radiations than MEFs isolated from WT littermates, most likely because the loss of HMGN1 impairs the removal of Cyclobutene pyrimidine dimers from the chromatin of UV irradiated cells. Taken together, these results indicate that loss of HMGN decreased the cellular ability to repair DNA damaged by UV radiation [27]. *Hmgn1^−/−^* mice and MEFs were also more sensitive to ionizing radiation than their wild-type littermates. Upon exposure to ionizing radiation, the tumor burden and mortality in *Hmgn1^−/−^* mice are higher as compared to their wild-type littermate mice [60]. The fibroblasts hypersensitivity to ionizing radiation is linked to altered G2-M checkpoint activation. Thus, in mice, loss of HMGN impairs repair of DNA damaged by either UV or ionizing radiation. Likewise, although DT40 chicken cells lacking either HMGN1, HMGN2 or both these variants survive [63], they are hypersensitive to UV irradiation. In these cells, loss of HMGNs increased the extent of G2-M checkpoint arrest and apoptosis [64]. An additional possible link between DNA damage and HMGN proteins is shown by a recent finding that the damage repressor protein Dsup of tardigrade *Ramazzottius varieornatus*, which binds to DNA and protects from hydroxy free radicals, exhibits sequence similarity with HMGN proteins [65].

In sum, the data support the possibility that the interaction of HMGN with chromatin optimizes DNA repair processes, while loss of HMGN variants decreases the efficiency of DNA repair, thereby increasing cell tumorigenicity. The mechanisms whereby HMGNs affect DNA repair processes are not fully understood; most likely, HMGNs affect DNA repair by modulating the interaction of DNA repair factors with chromatin, as was shown for ATM and for poly ADP-ribose polymerase-1 (PARP-1) [66,67,68].

## 9. Neurological Disorders

Several studies link altered HMGN1 to neurological disorders. Down syndrome, a common human genetic disorder affecting neural development, is caused by the presence of all, or part of an extra copy of chromosome 21 [69]. The *HMGN1* gene is located in a region of human chromosome 21 that is most frequently triplicated in Down syndrome, and tissue samples obtained from Down syndrome individuals overexpress HMGN1 protein [40,41]. A region in distal mouse chromosome 16 shows conserved linkage with human chromosome 21 and mouse trisomic in this region have been used as models for Down syndrome [70]. The mouse models for Down syndrome express approximately 1.5-fold higher levels of HMGN1 mRNA and protein, as compared to normal littermates [40,71,72]; however, it is not clear whether the neurological phenotypes seen in Down syndrome individuals are directly linked to elevated HMGN1 levels.

Alterations in the DNA binding protein methyl CpG-binding protein 2 (MeCP2) levels are known to affect neurodevelopmental disorders such as autism spectrum disorders, mental retardation, learning disabilities, Rett syndrome, repetitive behavior, hypotonia, and anxiety [73,74,75]. Targeted analysis of the Autism Genetic Resource Exchange genotype collection revealed a non-random distribution of genotypes within 500 kbp of HMGN1 in a region affecting its expression in families predisposed to autism spectrum disorders [43]. In mice, HMGN1 is a negative regulator of MeCP2 and analyses of genetically altered HMGN1 mice show that changes in HMGN1 levels are linked to abnormalities in activity and anxiety and social skills [43]. As indicated in the previous section, HMGN proteins regulate the expression of *Olig1* and *Olig2* and mice lacking HMGN1 and HMGN2 show decreased myelination and phenotypes that can be attributed to altered nerve impulse transmission [20]. Taken together, the available results suggest a role for HMGN variants in proper differentiation and function of neural cells and support the possibility that misexpression of these proteins could play a role in the etiology of neuronal disorders.

## 10. Immune Functions

Several studies reported the presence of autoantibodies against HMGN1 and HMGN2 in the serum of individuals with autoimmune diseases such as systemic lupus erythematosus, rheumatoid arthritis, mixed connective tissue disease, and scleroderma [76,77,78,79]. However, there is no clear evidence that HMGN variants play a role in the etiology of autoimmune disease. The antibodies seem to have been elicited by HMGN variants or chromatin fragments containing HMGN that are immunogenic due to cellular damage and nuclear breakdown in these diseases, to defects of the immunosurveillance system against autoantigens, or to “molecular mimicry” to an invading pathogen. Nevertheless, analyses of antibodies to HMGN in autoimmune disorders could provide clues to the etiology of these disorders [77].

Support for “molecular mimicry” between bacterial pathogens and HMGN proteins is provided by several studies reporting that HMGNs and HMGN-derived peptides show antimicrobial activity. The presence of HMGN2 in the cytoplasm and extracellular environment of mononuclear leukocytes highlights the function of HMGN2 as an antimicrobial effector. Full-length HMGN2 also displayed potential antimicrobial activity against *E. coli* ML-35p, *Pseudomonas aeruginosa* ATCC 27853, and *Candida albicans* ATCC 10231 and in vivo, inhibited the invasion of *K. pneumoniae* 03182 onto mouse lungs [80]. An antimicrobial peptide with the sequence PKRKAEGDAK, which is identical to the nucleosome binding domain of HMGN2 was isolated from human peripheral blood mononuclear leukocytes [81]. In several instances, the isolated alpha-helical domain peptide of HMGN2 has been shown to possess the same antimicrobial activity as full length HMGN2 [81]. In addition, HMGN2 seems to induce autophagy, a novel antimicrobial pathway, and protects bladder epithelial cells from infection by *E. coli* and other Gram-negative bacteria [82,83,84]. In agreement, recent studies indicated that deficiency in HMGN2 affected macrophage polarization and non-tuberculosis mycobacterial survival in macrophages [85]. The molecular mechanisms involved in the antibacterial activity of HMGN2 has not been studied in detail. Most likely, “molecular mimicry” between HMGN and bacterial components stimulate the immune system to mount cellular responses that counteract the deleterious effects of microbial infection.

Alarmins are endogenous molecules that stimulate defense responses including innate and adaptive immune systems. In this context, HMGN1 was shown to act as an alarmin by inducing maturation in human dendric cells via Toll-like receptor 4 (TLR-4) and by recruiting antigen-presenting cells (APCs) to the site of infection [86,87]. Furthermore, when administrated along with antigen, HMGN1 acts as an adjuvant and promoted specific immune response in WT but not in *Hmgn1^−/−^* mice [87]. Absence of tumor specific immune response in *Hmgn1^−/−^* mice and increased anti-tumor immune response, when vaccinated mice with DNA vector overexpressing HMGN1, provide further support the role of HMGN proteins in regulating immune cell function and response [88]. Intratumor delivery of HMGN1 and TLT7/8 synthetic agonist R848, along with a checkpoint inhibitor Cytoxan synergistically activates dendritic cells for optimal Th1 response and eradicated large established CT26 tumors in mice [89,90]. Moreover, low doses of HMGN1 with anti-CD4 depleting antibody promoted the expansion of anti-tumor CD8 T cells by rescuing them from exhaustion [91]. In a recent study on patients with head and neck carcinoma, it was observed that cytoplasmic localization and secretion of HMGN1 was associated with the recruitment of tumor infiltering lymphocytes [92]. A proteomics study in HMGN1 knockout revealed the role of HMGN1 in complement and coagulation cascade and interferon response in cancer cells, reinforcing the role of HMGN1 in regulating immune response [93]. These and additional studies suggest that HMGN1 can be used to increase the efficiency of certain antitumor treatments [89,94,95].

## 11. Perspective: Biological Function of the HMGN Protein Family

The biological function of HMGNs is a major unresolved question in chromatin biology.

Every vertebrate cell examined expresses HMGN proteins, the amino acid sequence of the proteins is conserved, the levels of these proteins correlate with specific developmental stages and several studies revealed that the protein binds specifically to chromatin, preferentially targeting regulatory sites such as enhancers and promoters [10,11]. Indeed, changes in HMGN levels can lead to altered cellular transcription profiles and a wide range of studies, including analyses of genetically altered mice, do correlate changes in HMGN levels with altered cellular phenotypes (Figure 1, Table 2 and Table 3).

Yet, in humans, not a single disease or developmental abnormality is directly attributed to or is strongly correlated with, either loss or decreased expression of an HMGN variant. The only direct correlation between changes in HMGN expression and a human phenotype is Down syndrome where an increase in HMGN1 has been shown to correlate with increased incidence of B-ALL [52]. Likewise, mice lacking HMGN are born and appear normal; however, detailed examinations reveal multiple phenotypes, especially when exposed to stress. A hypothetical explanation for these observations is that in humans, loss of HMGN expression leads to embryonic lethality; embryos with disrupted HMGN expression are not born. Mice lacking HMGN survive because under the controlled conditions of an animal facility loss, HMGN does not lead to embryonic lethality; presumably, under more stressful conditions found outside an animal facility, the mice would not be born or survive. In *Xenopus,* changes in HMGN levels lead to embryonic lethality, provided that the changes in HMGN levels are induced after midblastula transition [29].

A possible clue to molecular mechanisms regulating the biological function of the HMGN protein family comes from experiments showing that HMGNs bind to cell type-specific enhancers [11,96], that they do regulate the cell type-specific gene expression [18] and that they do stabilize, rather than determine cell identity, most likely by modulating the dynamic establishment and maintenance of a cell type-specific chromatin epigenetic landscape [11]. These findings suggest that cells do indeed survive without these proteins because they optimize and fine-tune rather than determine cell type-specific biological processes. The widespread occurrence and sequence conservation of HMGN proteins in vertebrate cells, and their association with cell type-specific chromatin regulatory sites, suggest that they do play important roles in safeguarding cell type-specific gene expression programs that optimize cellular functions necessary for proper development and prevention of disease.

## Figures and Tables

**Figure 1 ijms-21-00449-f001:**
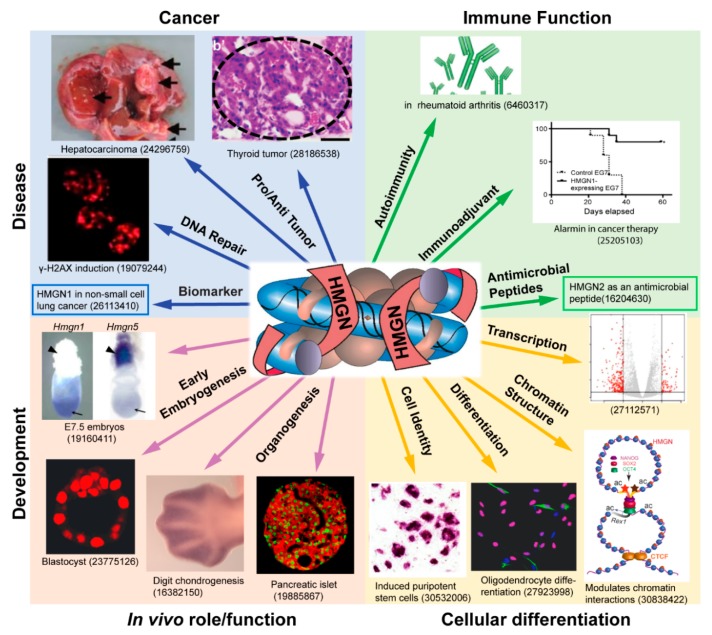
HMGN proteins in development and disease. Shown is a schematic illustration of the biological processes reported to be affected by members of the HMGN protein family. The center panel depicts a nucleosome complexed with two molecules of HMGN protein. The number shown below each panel is the PMID of an article that discusses a role of HMGN in a specific biological process. Not all the biological processes affected by HMGN are depicted. A comprehensive list of reference to articles on the biological functions of HMGN is presented in Table 2 and Table 3.

**Table 1 ijms-21-00449-t001:** Genetically modified high mobility group N (HMGN) mice *.

Gene	Modification	References
*Hmgn1*	Knockout	Birger, Y. et al., 2003
*Hmgn2*	Knockout	Deng, T. et al., 2015
*Hmgn2* ^#^	Knockout	Gao, X.L. et al., 2019
*Hmgn3*	Knockout	Ueda, T. et al., 2009
*Hmgn5*	Knockout	Ciappio, E.D. et al., 2014
*Hmgn1/Hmgn2*	Knockout	Deng, T. et al., 2015
Human *HMGN1*	Transgenic	Bustin, M. et al., 1995
Human *HMGN4*	Transgenic	Kugler, J. et al., 2017
*Hmgn5*	Transgenic	Furusawa, T. et al., 2016

* All mouse lines listed except the line denoted by ^#^ were generated in the author’s laboratory. Additional commercial resources for genetically altered mice may be available.

**Table 2 ijms-21-00449-t002:** **HMGN in Development**. Studies on the role of HMGN proteins in development and differentiation published in the English language. Reference given as PubMed Identifier (PMID).

HMGN	Species	Study Model	Summary	Reference (PMID)
HMGN1	Mouse	Myoblast differentiation	Overexpression of human HMGN1 inhibited the differentiation of mouse C2C12 myoblast cells into myotubes	8514795
HMGN1	Mouse	Transcription in MEFs	HMGN1 is a negative regulator of N-Cadherin gene. N-Cadherin was also higher in the early embryos of HMGN1 KO compared to WT	16279949
HMGN1	Mouse	Cornea differentiation and maturation	In corneal epithelium of Hmgn1 KO mice, corneal differentiation, maturation and gene expression were affected. In the wildtype mice, HMGN1 is colocalized with p63 and p63 expression was absent in the central region of the Hmgn1 KO cornea	16466397
HMGN1	Mouse	Chondrocyte differentiation	In developing limb bud, Hmgn1 expression is inversely correlated to Sox9 expression and it is down regulated during chondrogenesis	16382150
HMGN1	Mouse	Myogenesis and adipogenesis	Myogenesis or adipogenesis were induced in Hmgn1 knock down (KD) NIH3T3 cell line by overexpressing MyoD or C/EBPalpha. *Hmgn1* Knockdown cells did not show any differences when compared with control cells	16451822
HMGN1	Mouse	Digit formation	At E13.5 mouse autopod, interdigital expression of *hmgn1* mRNA is highest at the digit-interdigit junction and is greatly down-regulated in the Raldh2 KO mice	20034106
HMGN1	Mouse	embryonic stem cells (ESCs)	HMGN1 is strongly expressed in oocytes and throughout all preimplantation stages. Loss of HMGN1 reduces the DNase I hypersensitivity (DHS) at CpG island promoters in ES cells	23775126
HMGN1	Human	Transgenic mice	HMGN1 is negative regulator for methyl CpG-binding protein 2. Hence, mice overexpressing HMGN1 displayed abnormalities in activity and anxiety and to social deficits	22009741
HMGN1	Mouse	Down syndrome model mouse.	HMGN1 is overexpressed in Down syndrome and in all down syndrome model mice	2140193
				16635258
HMGN2	Mouse	Kidney differentiation	During metanephric kidney development, HMGN2 expression is highly linked to induction and differentiation in kidney organogenesis	11683498
HMGN1 HMGN2	Mouse	Myoblast differentiation	During myogenesis the level of *Hmgn1* and *Hmgn2* mRNA were decreased and this downregulation is a differentiation related event	1689723
HMGN1 HMGN2	Chicken	Erythroid cells	Erythroid cells from 5-day chicken embryos contain higher level of *Hmgn* mRNAs than cells from 14-day embryos. Change from primitive to definitive erythroid lineage during embryogenesis is accompanied by a change in the expression of HMGN proteins.	1993650
HMGN1 HMGN2	Rat Human	Osteoblast differentiation and monocyte differentiation	In both primary cultures of calvarial-derived rat osteoblasts and human leukemia HL-60 cells, HMGN1 and HMGN2 are preferentially expressed in the proliferating state, and the levels are down-regulated during differentiation.	8496248
HMGN1 HMGN2	Mouse	Early embryonic development	HMGN1 and HMGN2 were detected throughout oogenesis and embryonic transcripts accumulate beyond the two-cell stage. Depletion of HMGNs in one-cell or two-cell mouse embryos delays subsequent progression.	11133167
HMGN1 HMGN2	Xenopus	Early embryonic development	Depletion of *Hmgn1* or *Hmgn2* resulted in developmental defects after mid blastula transition (MBT) in Xenopus embryos, and re-expression of HMGN2 rescued these phenotypes	14568106
HMGN1 HMGN2	Bovine	Maternal-zygotic transition	During bovine embryo early stage development, HMGN1 and HMGN2 are expressed in oocytes but are downregulated after fertilization and almost disappear by the 8-cell stage	14960490
HMGN1 HMGN2	Mouse	Hair follicle cycle	*Hmgn* mRNAs was expressed at the onset of hair follicle development and during the adult hair follicle cycle	19303948
HMGN1 HMGN2	Mouse	MEF, mouse behavior	Genome wide mapping of DHSs sites in MEFs from HMGN1 KO, HMGN2 KO and HMGN DKO revealed that loss of both, but not single HMGN variant leads to significant remodeling of DHS especially at enhancer regions	26156321
HMGN1 HMGN2	Mouse	B lymphocyte	Loss of HMGN1 and HMGN2 proteins affect gene expression by modulating chromatin regulatory sites during activation of naïve B cells	27112571
HMGN1 HMGN2	Mouse	Oligodendrocyte differentiation	Loss of HMGNs affect the expression of OLIG2 and OLIG1, thereby affecting oligodendrocyte development and nerve myelination, and mouse behavior	27923998
HMGN1 HMGN2	Mouse	MEFs, induced pluripotent stem cells (iPSCs)	MEFs from Hmgn1/2 DKO mice showed enhanced rate of iPSCs induction by OSKM factors, and into neurons induced by ASCL1	30532006
HMGN1 HMGN2 HMGN3	Mouse	Eye development	HMGN1, HMGN2 and HMGN3 displayed dynamic and specific expression patterns during ocular development (E10.5, E13.5 and newborn) and adult eye	18502697
HMGN1 HMGN2 HMGN3	Mouse	Astrocyte differentiation	Ectopic expression of either HMGN1, HMGN2 or HMGN3 in neural precursor cells (NPCs) or in late embryonic neocortex promoted the generation of astrocytes while knockdown suppressed astrocyte differentiation.	25069414
HMGN2	Bovine	Early embryonic development	An in vitro fertilization study of bovine embryos revealed that embryos with enhanced levels of HMGN2 failed to develop into blastocyst due to the altered chromatin remodeling which might be caused by hyperacetylation of Histone H3 at lysine 14 residue	17712799
HMGN2	Mouse	Tooth development	HMGN2 binds to Pitx2 and inhibits Pitx2 mediated Amelogenin expression during tooth development	23975681
HMGN2	Mouse	Corticogenesis (Cerebral cortex development)	*Hmgn2* KO mice exhibited microcephalus phenotype with reduced cortical surface area and almost normal radial corticogenesis	31699896
HMGN3	Xenopus	Anuran metamorphosis	*Hmgn3* expression is upregulated during thyroid hormone dependent anuran metamorphosis.	11921340
HMGN3	Mouse	Retina	HMGN3 is highly expressed in glia cells and eye. HMGN3 binds to glycine transporter 1 (Glyt1) gene and upregulates its expression.	15082770
HMGN3	Mouse	Pancreas (beta cells)	Loss of HMGN3 abrogates glucose stimulated insulin secretion and HMGN3 KO mice possess diabetic phenotype. HMGN3 along with PDX1regulates Glu2 gene	19651901
HMGN3	Bovine	Early embryonic development	*Hmgn3* mRNA was expressed at similar levels in bovine matured oocytes and 2–4-cell embryos, but expression was higher in 8-16-cell embryos, morulae and blastocysts	19393058
HMGN3	Mouse	Pancreas (alpha cells)	HMGN3 is highly expressed in pancreatic alpha cells, and that Hmgn3 KO mice displayed reduced glucagon levels in their blood	19885867
HMGN3	Mouse	Decidualization	HMGN3 is highly expressed in decidua and decidualizing stromal cells, and also in the artificially stimulated decidualization	26112184
HMGN5	Mouse	Placenta development	*Hmgn5* mRNA is strongly expressed in ectoplacental cone in E7.5 embryo and its expression retained in developing placenta and in trophoblast giant cells	19160411
HMGN5	Mouse	Metabolomics	Metabolomic analysis of liver extracts and urine from *Hmgn5* KO revealed that loss of HMGN5 leads to downregulation of glutathione peroxidase 6 (Gpx6) and hexokinase 1 (Hk1), thereby affects glutathione metabolism	24392144
HMGN5	Mouse	Heart development	Transgenic mice overexpressing HMGN5 (both ubiquitous or heart specific) are normal at birth but develop hypertrophic heart with large cardiomyocytes, deformed nuclei and disrupted lamina and die of cardiac malfunction	25609380

**Table 3 ijms-21-00449-t003:** **HMGN in Disease**. Studies on the role of HMGN proteins in different diseases published in English language. Reference given as PMID.

HMGN	Study Model	Disease	Role/Function	Summary	Reference (PMID)
HMGN1	Mice	Cancer	Immunoadjuvant	Augments antitumor immunity of alpha-fetoprotein expressing lentiviral vaccines in hepatocellular carcinoma mice	31281528
HMGN1	Mice	Cancer	Immunoadjuvant	Treatment with HMGN1 and anti-CD4 depleting antibody reverses T cell exhaustion and exerts robust anti-tumor effects in mice	30696484
HMGN1	Cell culture	Cancer	Immunoadjuvant	HMGN1 promotes dendritic cells (DC) recruitment through interacting with a Gαi protein-coupled receptor and activates DCs predominantly through triggering toll like receptor 4 (TLR4)	29503123
HMGN1	Mice	Cancer	Immunoadjuvant	Intratumoral delivery of HMGN1 and R848 along with cytoxan (TheraVac) eradicated large established CT26, Renca and EG7 tumors in mice	29079801
HMGN1	Mice	Cancer	Immunoadjuvant	Intratumoral injections cured mice harboring large established subcutaneous Hepa1-6 hepatomas	28881713
HMGN1	Mice	Cancer	Immunoadjuvant	HMGN1 contributes to antitumor immunity	25205103
HMGN1	Mice	Cancer	Immunoadjuvant	HMGN1 induced DC maturation via TLR4, recruitment of antigen presenting cells at sites of injection and promoted antigen-specific immune response upon co-administration with antigens	22184635
HMGN1	Mice	Cancer	Immunoadjuvant	DNA(TA) encapsulated within peptide hydrogels encodes for a melanoma-specific gp100 antigen fused to the HMGN1 enhanced immunostimulatory effects in tumor mice model	25890750
HMGN1	Mice	Cancer	Protumor activity	Down syndrome associated triplication of a 21q22 region contributes to B cell proliferation with increased risk of B cell acute lymphoblastic leukemia through HMGN1 overexpression	24747640
HMGN1	Cell culture	Cancer	Protumor activity	High expression of HMGN1 in highly metastatic cells of MDA-MB-435 suggests the association of HMGN1 with metastasis of breast cancer cells	16979889
HMGN1	Patient’s studies	Cancer	Protumor activity	Endoplasmic reticulum stress induces secretion of HMGN1 and is associated with tumor-infiltrating lymphocytes in triple-negative breast cancer	27494867
HMGN1	Patient’s studies	Cancer	Protumor activity	High cytoplasmic expression of HMGN1 was associated with a high histological grade, high levels of tumor infiltering lymphocytes (TILs) in HER2-positive breast cancer tissues	26445971
HMGN1	Patient’s studies	Cancer	Biomarker	Patients with high serum HMGN1 had a poorer overall survival after curative pneumonectomy in non-small cell lung cancer	26113410
HMGN1	Cell culture	Cancer	Transcriptional regulator	HMGN1 and HMGN2 are synthesized throughout the S-phase and persists through the cell cycle in Burkitt’s lymphoma	8670211
HMGN1	Mice	Cancer	Antitumor activity	Loss of HMGN1 accelerates the progression of N-nitrosodiethylamine-induced hepatocarcinogenesis	24296759
HMGN1	Mice	Cancer	Antitumor activity	Loss of HMGN1 increases the tumorigenicity and sensitivity to Ionizing radiations	16061652
HMGN1	Patient’s studies	Down Syndrome	Transcriptional regulator	Gene dosage studies in cultured cells and brain tissue samples obtained from Down syndrome patients suggest that HMGN1 as contributing factor in the etiology of Down syndrome	1825298
HMGN1	Mice	Down Syndrome	Gene locus	HMGN1 maps to the 21q22.3 in humans, the region associated with the pathogenesis of Down syndrome, and to chromosome 16 in mice	2140193
HMGN1	Mice	Down Syndrome	Transcriptional regulator	Overexpression of the HMGN1 (encoded on chr21q22) reiterates transcriptional changes seen with triplication of Down syndrome critical region on distal chromosome 21	30428356
HMGN1	Mice	Diabetic nephropathy	Transcriptional regulator	HMGN1 expression correlates with epithelial mesenchymal transition, renal fibrosis in diabetic nephropathy mice	31203283
HMGN1	Mice	Diabetic nephropathy	Transcriptional regulator	Insulin growth factor 1 receptor inhibitor attenuates diabetic nephropathy by suppressing HMGN1-TLR4 pathway	29384065
HMGN1	Rats	Amnesia	Biomarker	Increased HMGN1 phosphorylation by protein kinase CK2 decreases amnesia in aged rats	14635996
HMGN1	Patient’s studies	Systemic sclerosis	Immunogen in autoimmunity	Autoantibodies against HMGN1 and HMGN2 were found in systemic sclerosis patients’ sera	7869312
HMGN1	Patient’s studies	Drug induced lupus	Immunogen in autoimmunity	Autoantibodies against HMGN1 and HMGN2 were found in drug induced lupus patients	7907477
HMGN1	Patient’s studies	Primary pulmonary hypertension	Immunogen in autoimmunity	Autoantibodies against HMGN1 were found in sera from primary pulmonary hypertension patients	8918589
HMGN1	Mice	Immune Tolerance	TLR-4 agonist	HMGN1 induces innate immune tolerance in a TLR4 dependent Sirtuin-1 mediated deacetylation in human blood peripheral mononuclear cells	29593748
HMGN2	Mice	Cancer	Antitumor activity	The tumor volumes in SaO2 and U2-OS osteocarcinoma subcutaneous nude mouse models treated with overexpressing HMGN2 lentivirus were significantly decreased	25530340
HMGN2	Mice	Cancer	Antitumor activity	Overexpression of HMGN2 significantly decreased migration capacity, increased apoptosis of MCF7 breast cancer cells both in vitro and in vivo	30655878
HMGN2	Cell culture	Cancer	Protumor activity	HMGN2 was increased in tumor cell line MDA-MB-468 and also in metastatic oral squamous cell carcinoma tissues	29541221
HMGN2	Cell culture	Cancer	Antitumor activity	Isopentenyl pyrophosphate activated *γδ* T cells expresses high level of HMGN2 which contributes to antitumor activity	29401165
HMGN2	Cell culture	Cancer	Antitumor activity	Overexpression of HMGN2 in by lentivirus, inhibits the growth and migration of osteosarcoma cells	25530340
HMGN2	Cell culture	Cancer	Signal transducer	Surface expressed of HMGN2 on leukemic B cells binds to autoantigen erythrocyte protein band 3 and contributes to initiation of autoimmune hemolytic anemia	25156469
HMGN2	Cell culture	Cancer	Antitumor activity	Phytohemagglutinin activated CD8 T cells releases high level of HMGN2 and kill tumor cells	25060707
HMGN2	Cell culture	Cancer	Protumor activity	Prolactin induced H1 disassociation mediated by HMGN2 induce STAT5 recruitment and breast cancer pathogenesis	28035005
HMGN2	Patient studies	Cancer	Protumor activity	Expression of HMGN2 was higher in human leukemia cells with chronic myelogenic leukemia	8528142
HMGN2	Cell culture	Cancer	Protumor activity	HDAC6 mediated deacetylation of HMGN2 enhances Stat5a transcriptional activity, thereby regulating prolactin-induced gene transcription and breast cancer growth	27358110
HMGN2	Cell culture	Cancer	Biomarker	HMGN1 and HMGN2 are synthesized throughout the S-phase and persists through the cell cycle in Burkitt’s lymphoma	8670211
HMGN2	Cell culture	Cancer	Antitumor activity	HMGN2 is a direct target of MicroRNA-23a which plays crucial role in epithelial mesenchymal transition (EMT) in lung cancer cells suggests its role	23437179
HMGN2	Mice	Cancer	Antitumor activity	HMGN2 exerts inhibitory effect on tumors in human tongue carcinoma transplanted nude mice	24665631
HMGN2	Mice	Cancer	Antitumor activity	HMGN2 treatment inhibited the growth, increased apoptosis of Tca8113 cells in vitro and in vivo	24348831
HMGN2	Cell culture	Cancer	Antimicrobial activity	31-aa peptide (F3) in the HMGN2 sequence homed to HL-60 and MDA-MB-435 tumors upon I.v. injection	12032302
HMGN2	Patient studies	Cancer	Immunogen in autoimmunity	High throughput screening of autoantibodies in ovarian cancer ascitic fluids revealed HMGN2 as an antigen	31428516
HMGN2	Cell culture	Infection	Transcriptional regulator	HMGN2 regulates the expression of antimicrobial peptides in Uropathogenic Escherichia coli J96-infected bladder epithelial cell monolayer	29549670
HMGN2	Cell culture	Infection	Antimicrobial activity	HMGN2 displayed anti-bacterial and anti-biofilm activity against Escherichia coli K12	28471113
HMGN2	Cell culture	Infection	Antimicrobial activity	HMGN2 inhibits Pseudomonas aeruginosa internalization in A549 cells	28408162
HMGN2	Cell culture	Infection	Antimicrobial activity	HMGN2 protects bladder epithelial cells from Klebsiella invasion	21720014
HMGN2	Cell culture	Infection	Transcriptional regulator	Knockdown of HMGN2 affected the survival of non-tuberculosis mycobacteria in macrophage	31596045
HMGN2	Cell culture	Infection	Transcriptional regulator	HMGN2 is a target of miR-155, which is downregulated in Klebsiella pneumoniae infection suggested the role in controlling infection	27534887
HMGN2	Cell culture	Infection	Transcriptional regulator	Knockdown of HMGN2 increases the internalization of Klebsiella pneumoniae by respiratory epithelial cells	27460641
HMGN2	Mice	Infection	Antimicrobial activity	HMGN2 is crucial for LPS induced expression of murine β-defensin-3 and -4 in mice	21594618
HMGN2	Cell culture	Infection	Antimicrobial activity	HMGN2 is crucial for LPS induced expression of murine β-defensin-3 and -4 in A549 cells	21518253
HMGN2	Mice	Infection	Antimicrobial activity	HMGN2 inhibits the invasion of Klebsiella pneumoniae into mouse lungs in vivo	25760831
HMGN2	Cell culture	Infection	Antimicrobial activity	HMGN2 inhibited hepatitis B virus infection and replication in HepG2.2.15 cell line	19150374
HMGN2	Cell culture	Infection	Antimicrobial activity	HMGN2 inhibited internalization of Klebsiella pneumoniae into cultured bladder epithelial cells	21778192
HMGN2	Cell culture	Infection	Antimicrobial activity	N-terminal amino sequence was PKRKAEGDAK of HMGN2 is a novel antimicrobial peptide	16204630
HMGN2	Cell culture	Infection	Antimicrobial activity	HMGN2 showed potent antimicrobial activity against E coli ML-35p, P aeruginosa ATCC 27853	16115376
HMGN2	Patient’s studies	Systemic lupus erythromatosus	Immunogen in autoimmunity	Autoantibodies against HMGN2 were found in systemic lupus erythromatosis (SLE) patients	8037838
HMGN2	Patient’s studies	Systemic lupus erythromatosus	Immunogen in autoimmunity	Autoantibodies to HMGN2 were found in different autoimmune diseases and are more frequent in Systemic lupus erythromatosus	8318042
HMGN2	Patient’s studies	Systemic lupus erythromatosus	Immunogen in autoimmunity	Autoantibodies to HMGN2 were present in serums from patients with systemic lupus erythematosus (SLE), rheumatoid arthritis (RA), and mixed connective tissue disease (MCTD)	6460317
HMGN2	Patient’s studies	Drug induced lupus	Immunogen in autoimmunity	Autoantibodies against HMGN1 and HMGN2 were found in drug induced lupus patients	7907477
HMGN2	Patient’s studies	Juvenile Rheumatoid Arthritis	Immunogen in autoimmunity	Autoantibodies to HMGN2 were detected in ANA-positive patients with periarticular-onset Juvenile Rheumatoid Arthritis	7517709
HMGN2	Patient’s studies	Juvenile Rheumatoid Arthritis	Immunogen in autoimmunity	Autoantibodies to HMGN2 were detected in patients with Juvenile Rheumatoid Arthritis	1567496
HMGN2	Patient’s studies	Periodontitis	Biomarker	HMGN2 in saliva of periodontitis patients was higher especially with severe periodontitis	28591948
HMGN2	Mice	Microcephaly	Transcriptional regulator	HMGN2 protects against Microcephaly by maintaining chromatin accessibility during corticogenesis	31699896
HMGN3	Mice	Diabetes	Anti-diabetic	Hmgn3^-/-^ mice which have a mild diabetic phenotype, with reduced glucagon levels in the blood	19885867
HMGN3	Mice	Diabetes	Anti-diabetic	In mice, loss of HMGN3 blunt glucose-stimulated insulin secretion and have a diabetic phenotype. HMGN3 regulates Glut4 gene in pancreatic beta cells	19651901
HMGN4	Patient’s studies	Cancer	Biomarker	Patients with high expression of HMGN4 have high grade tumors, shorter overall survival in hepatocellular carcinoma	30272824
HMGN4	Mice	Cancer	Protumor activity	Mouse and human cells overexpressing HMGN4 displayed increased tumorigenicity in thyroid tumor mouse model	28186538
HMGN5	Cell culture	Cancer	Protumor activity	microRNA-183-3p directly inhibits the expression of HMGN5 and prostate cancer progression	31314587
HMGN5	Cell culture	Cancer	Protumor activity	HMGN5 expression was upregulated in pancreatic ductal adenocarcinoma tissues and cell lines.	30128022
HMGN5	Cell culture	Cancer	Protumor activity	miR-488 inhibits cell growth and metastasis in renal cell carcinoma by abrogating HMGN5 expression	29713189
HMGN5	Cell culture	Cancer	Protumor activity	HMGN5 expression was significantly upregulated in bladder transitional cell carcinoma tissues	29509244
HMGN5	Cell culture	Cancer	Protumor activity	Silencing HMGN5 promotes chemosensitivity of human bladder cancer cells to cisplatin	29163683
HMGN5	Cell culture	Cancer	Protumor activity	HMGN5 was significantly upregulated in esophageal squamous cell carcinoma cells. Silencing of HMGN5 significantly inhibited cell growth and induced cell apoptosis of ESCC cells	28914995
HMGN5	Cell culture	Cancer	Protumor activity	microRNA-495 inhibits the proliferation and invasion and induces the apoptosis of osteosarcoma cells by inhibiting HMGN5	28627703
HMGN5	Cell culture	Cancer	Protumor activity	MicroRNA-140-5p expression enhances osteosarcoma chemoresistance by targeting HMGN5 and autophagy	28341864
HMGN5	Cell culture	Cancer	Protumor activity	MicroRNA-409-3p decreases glioma cell invasion and proliferation by targeting HMGN5	28109076
HMGN5	Cell culture	Cancer	Protumor activity	microRNA-340 inhibits tumorigenic potential of prostate cancer cells by abrogating HMGN5 expression	26394192
HMGN5	Cell culture	Cancer	Protumor activity	HMGN5 silencing enhances apoptosis, suppresses invasion and increases chemosensitivity to temozolomide in meningiomas	26315299
HMGN5	Cell culture	Cancer	Protumor activity	HMGN5 silencing suppresses the viability and invasion of human urothelial bladder cancer 5637 cells	25796505
HMGN5	Cell culture	Cancer	Protumor activity	HMGN5 is over expressed in prostate cancer cells and activates MAPK and its expression is correlated with chemosensitivity	25572120
HMGN5	Cell culture	Cancer	Protumor activity	HMGN5 is highly expressed in breast cancer cells and prompts proliferation and invasion	25315189
HMGN5	Cell culture	Cancer	Protumor activity	HMGN5 silencing in cells exhibit increased apoptosis rate and decreased colonogenic survival in response to 2–8 Gy ionizing radiation	25307178
HMGN5	Cell culture	Cancer	Protumor activity	HMGN5 was positively correlated with pathologic staging and TNM staging of osteosarcoma	24687550
HMGN5	Cell culture	Cancer	Protumor activity	HMGN5 increases drug resistance in osteosarcoma cell lines U-2OS and MG63 through upregulating autophagy	24664583
HMGN5	Cell culture	Cancer	Protumor activity	Silencing of HMGN5 induces apoptosis through the modulation of a mitochondrial pathway and Bcl-2 family proteins in prostate cancer cells	22504871
HMGN5	Cell culture	Cancer	Protumor activity	Knockdown of HMGN5 slowed down the cell cycle at the G0/G1 phase in human lung cancer cell lines A549 and H1299	22994738
HMGN5	Cell culture	Cancer	Protumor activity	HMGN5 silencing inhibits the growth and invasion of clear cell renal cell carcinoma cells and its expression is associated with tumor grade	22420896
HMGN5	Cell culture	Cancer	Protumor activity	HMGN5 promotes the viability of bladder cancer cells through increased cell proliferation, invasion ability of metastatic bladder cancer cells through the upregulation of MMP-9	21695596
HMGN5	Cell culture	Cancer	Protumor activity	HMGN5 knockdown causes cell cycle arrest in human glioma U251 and U87 cells and it is highly expressed in high grade and low-grade tumor tissues	21373965
HMGN5	Cell culture	Cancer	Protumor activity	HMGN5 knockdown causes cell cycle arrest in prostate cancer cell line DU145	20531280
HMGN5	Cell culture	Cancer	Protumor activity	HMGN5 knockdown inhibits proliferation and tumorigenicity of BGC823 and SGC7901 gastric cancer cell lines	30854121
HMGN5	Cell culture	Cancer	Protumor activity	MicroRNA-186 suppresses the growth and metastasis of bladder cancer by targeting HMGN5	26290438
HMGN5	Cell culture	Cancer	Protumor activity	HMGN5 knockdown inhibits proliferation and tumorogenecity of BGC823 and SGC7901 gastric cancer cell lines	30854121
HMGN5	Cell culture	Cancer	Protumor activity	MicroRNA-186 suppresses the growth and metastasis of bladder cancer by targeting HMGN5	26290438

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
