# Peer review of "Biological Functions of HMGN Chromosomal Proteins"

_ijms, 2020, doi:10.3390/ijms21020449_

Round 1
Reviewer 1 Report
This manuscript is an excellent review of the literature on the biological functions of HMGN chromosomal proteins. The organization is outstanding and the tables provide a thoughtful summary and will be an exceptional resource for other investigators.
Author Response
Answer to reviewer 1. We are pleased that this reviewer felt that the review is of high quality and that it will serve as a valuable resource. We double checked that the manuscript is free of any misspellings
Reviewer 2 Report
Nanduri et al reviewed HMGN biological functions comprehensively, especially the tables 2 & 3 compiled very long lists. It would be more attractive if this information can be extracted and summarised in a concise way. Table 1 is not very informative, it would be better if summarised phenotypes can be included.
Author Response
Response: We are pleased that also this reviewer found our manuscript well organized and significant. The aim of the review is to provide an UpToDate summary of the research on the biological function of HMGN. In this respect, Table 1 gives information on the currently available mouse models that can be used for these studies. In the text we have already extracted and summarized the information obtained with these the mice. In addition, in the text we provide a web site and additional information as to where the phenotypes are described in more detail. Likewise, the text summarizes the most important information obtained from the experiments listed in Tables 2 and 3. The tables are aimed to provide information that will facilitate easy access to all the studies that have perform on HMGN and support the narrative of the review.
Also we have spell checked the manuscript to make sure that it is free of grammatical errors.